# Machine learning enables improved runtime and precision for bio-loggers on seabirds

Joseph Korpela[1], Hirokazu Suzuki[2], Sakiko Matsumoto[2], Yuichi Mizutani[2], Masaki Samejima[1],
Takuya Maekawa [1✉], Junichi Nakai[3] & Ken Yoda [2]

Unravelling the secrets of wild animals is one of the biggest challenges in ecology, with bio-logging (i.e., the use of animal-borne loggers or bio-loggers) playing a pivotal role in tackling this challenge. Bio-logging allows us to observe many aspects of animals' lives, including their behaviours, physiology, social interactions, and external environment. However, bio-loggers have short runtimes when collecting data from resource-intensive (high-cost) sensors. This study proposes using AI on board video-loggers in order to use low-cost sensors (e.g., accelerometers) to automatically detect and record complex target behaviours that are of interest, reserving their devices' limited resources for just those moments. We demonstrate our method on bio-loggers attached to seabirds including gulls and shearwaters, where it captured target videos with 15 times the precision of a baseline periodic-sampling method. Our work will provide motivation for more widespread adoption of AI in bio-loggers, helping us to shed light onto until now hidden aspects of animals' lives.

[1] Graduate School of Information Science and Technology, Osaka University, Suita, Osaka 565-0871, Japan. [2] Graduate School of Environmental Studies, Nagoya University, Nagoya, Aichi 464-8601, Japan. [3] Graduate School of Dentistry, Tohoku University, Sendai, Miyagi 980-8575, Japan. ✉email: maekawa@ist.osaka-u.ac.jp

Animal-borne data loggers, i.e., bio-loggers, have revolutionised the study of animal behaviour in the animals' natural environments, allowing researchers to gain great insights into various aspects of the animals' lives, such as their social interactions and interactions with their environments[1–3]. Although there have been extraordinary improvements in the sensors and storage capacities of bio-loggers since the first logger was attached to a Weddell seal[4–9], their data collection strategies have remained relatively simple: record data continuously, record data in bursts (e.g., periodic sampling), or use manually determined thresholds to detect basic collection criteria such as a minimum depth, acceleration threshold, or illumination level[10–17]. However, these data collection strategies fall short when attempting to collect data using resource-intensive sensors (e.g., video cameras) from specific animal behaviours, as they tend to deplete all of the bio-loggers' resources on non-target behaviours[18,19]. This is especially true when working with animals such as birds, since the mass of a bio-logger is restricted to a small fraction of the bird's mass[8], e.g., the video bio-loggers used in this study weighed as little as 27 g ("Methods"), which greatly restricts the maximum battery capacity.

In this study, we propose the concept of AI-assisted bio-loggers, which we will refer to as AI on Animals (AIoA), that can use low-cost (i.e., non-resource-intensive) sensors to automatically detect behaviours of interest in real time, allowing them to conditionally activate high-cost (i.e., resource-intensive) sensors to target those behaviours. Using AIoA, these bio-loggers can limit their use of high-cost sensors to times when they are most likely to capture the target behaviour, increasing their chances of success by extending their runtime (e.g., from 2 h when continuously recording video to up to 20 h when using AIoA). To the best of our knowledge, the bio-loggers used in this study are the first AI-enabled bio-loggers ever to have been used in the wild. They include an integrated video camera (high-cost sensor) along with several low-cost sensors including an accelerometer and GPS unit.

## Results

**AI-assisted bio-logging**. Figure 1 shows an example of how AIoA can be used to capture videos of foraging behaviour when used on a bio-logger attached to a black-tailed gull (*Larus crassirostris*). In this example, the bio-logger pictured in Fig. 1a can detect foraging activity using acceleration data (Fig. 1c) in order to extend its runtime by only recording video during periods of that activity, which are indicated by the green segments shown in Fig. 1d, allowing it to capture target behaviours such as those pictured in Fig. 1e, f (see also Supplementary Movies 1 and 2). See Methods for a description of the machine learning algorithm used when recognising behaviours on board the bio-loggers.

**Experiment of black-tailed gulls**. We evaluated the effectiveness of our method by using AIoA-based camera control on board ten bio-loggers (Supplementary Fig. 1) that were attached to black-tailed gulls from a colony located on Kabushima Island near Hachinohe City, Japan[18], with the AI trained to detect possible foraging behaviour based on acceleration data. Figure 2 illustrates the results based on an evaluation of the videos by the ecologists participating in this study (see also Supplementary Movie 3). Of the 95 videos collected using the naive method, only 2 contained any target behaviour (2 possible foraging) while 93 videos contained non-target behaviour (11 flying and 82 stationary), giving a precision of about 0.02. In contrast, of the 184 videos collected using the proposed method, 55 contained target behaviour (4 confirmed foraging and 51 possible foraging) and 129 contained non-target behaviour (86 flying and 43 stationary), giving the proposed method a precision of about 0.30, which is about 15

times the expected precision of random sampling given that the target comprises only 1.6% of the dataset (see Fig. 2d and the following investigation). This high precision also illustrates the robustness of the proposed method, considering that it was achieved in the wild using AI trained from data collected from different hardware (Axy-Trek bio-loggers; see "Methods") and in some cases different positions on the animals (the back vs. the abdomen). A two-sided Fisher's exact test was performed to compare the results for the proposed method and the naive method ($p = 1.618 \times 10^{-9}$, odds ratio = 19.694, 95% confidence interval: [4.975–170.549]).

Along with the video-based evaluation, we also analysed the results for the black-tailed gulls by first fully labelling the low-cost sensor data (i.e., accelerometer data) collected by the bio-loggers and then computing the precision, recall, and *f*-measure for 1-s windows of sensor data, using the presence or absence of video data to determine the classifier output. Based on this full labelling of the data, the proposed method achieved a precision of 0.27, a recall of 0.56, and an *f*-measure of 0.37, while the naive method achieved a precision of 0.00, a recall of 0.00 and an *f*-measure of 0.00. Using this full labelling of the data, we were also able to compute an estimated distribution of the behaviours in the collected data and found that the target behaviour (foraging) comprised only about 1.62% of the 116 total hours of data collected, with 9.96% corresponding to flying behaviour and the remaining 88.43% corresponding to stationary behaviour. Thus, the proposed method was able to capture about half of the target behaviour and well outperformed the expected precision of random sampling when the target comprises only 1.62% of the dataset.

**Experiment of streaked shearwaters**. Along with the above evaluation that used acceleration data to train the machine learning models, we also evaluated the proposed method when training the models with GPS data. The bio-loggers were attached to streaked shearwaters (*Calonectris leucomelas*) from a colony located on Awashima Island in Niigata Prefecture, Japan. The AI on these loggers was trained to detect area restricted search (ARS), i.e., local flight activity in which the birds respond to patchily distributed prey, based on features extracted from the GPS data such as the distance travelled and average speed (Supplementary Fig. 2). Along with the AIoA-based loggers, we also deployed one naive logger that controlled the camera by simply turning it on once every 30 min (periodic sampling). Both the naive and AIoA-based loggers recorded 5-min duration videos. Supplementary Fig. 3 shows an overview of these results based on an evaluation of the videos by the ecologists participating in the study. See also Supplementary Movie 4. Of the 15 videos collected using the naive method, only 1 contained target behaviour (ARS) while 14 videos contained only non-target behaviour (1 transit and 13 stationary), giving a precision of about 0.07. In contrast, of the 32 videos collected using the proposed method, 19 contained target behaviour (ARS) and 13 contained non-target behaviour (7 transit and 6 stationary), giving the proposed method a precision of about 0.59. A two-sided Fisher's exact test was performed to compare the results for the proposed method and the naive method ($p = 1.085 \times 10^{-3}$, odds ratio = 19.277, 95% confidence interval: [2.390–904.847]).

Along with the video-based evaluation, we also fully labelled the low-cost sensor data (i.e., GPS data) collected by the loggers and computed the precision, recall, and *f*-measure for 1-s windows of sensor data, with the presence or absence of video again used in lieu of classifier output for each 1-s window. Based on this evaluation, the proposed method achieved a precision of 0.65, a recall of 0.14, and an *f*-measure of 0.23, while the naive

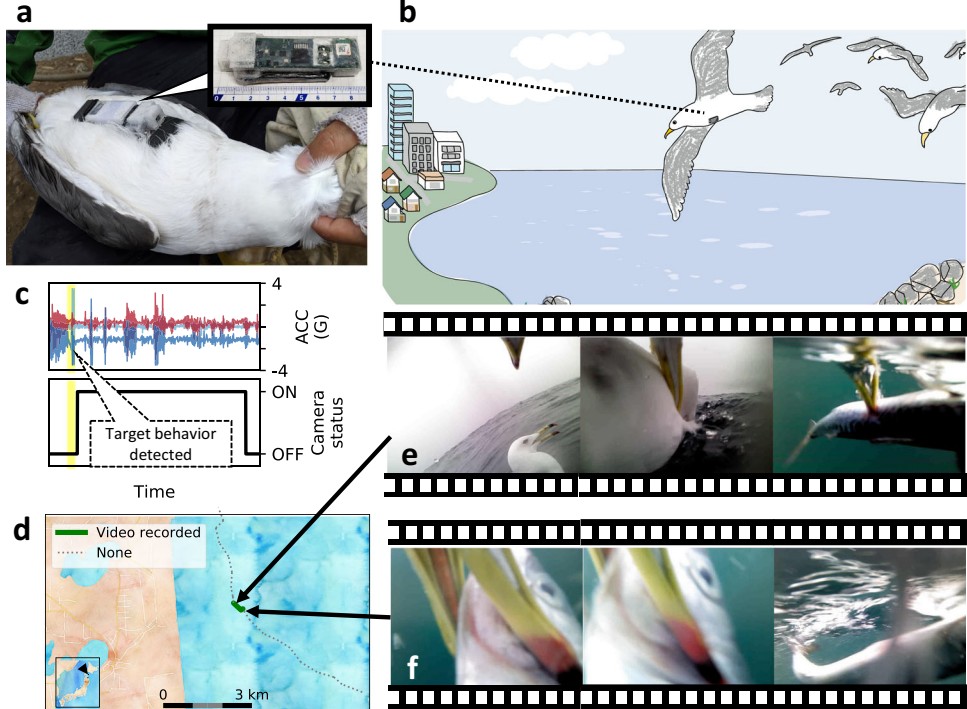

**Fig. 1 Example use of our AI-assisted bio-logger in the field. a** A bio-logger attached to the abdomen of a black-tailed gull. The bird is shown with its abdomen facing upward, with the bio-logger's camera lens facing towards the bird's head. **b** After attaching the bio-logger, the bird is then released to roam freely in its natural environment. **c** An accelerometer (low-cost sensor) can be used to control the bio-logger's video camera (high-cost sensor) by detecting body movements that are characteristic of the target behaviour (e.g., diving), activating the camera upon detection. **d** A GPS track captured by the bio-logger as the bird was flying off the coast of Aomori Prefecture, Japan. The portion of the track highlighted in green shows where videos (**e**) and (**f**) were captured, i.e., predicted target behaviour. (Map tiles by Stamen Design, under CC BY 3.0. Data by OpenStreetMap, under CC BY SA.) **e** Frames taken from a video captured using AI on Animals (AIoA) that show intraspecific kleptoparasitism by a black-tailed gull. **f** Frames taken from a video captured using AIoA that show a black-tailed gull catching a fish. Supplementary Data 1 provides source data of this figure.

method achieved a precision of 0.13, a recall of 0.13 and an *f*-measure of 0.13. We also computed an estimated distribution of the behaviours in the dataset based on this full labelling, with the target behaviour (ARS) found to constitute about 23.20% of the approximately 59 h of data collected while transit behaviour constituted about 34.36% and stationary behaviour constituted about 42.43%. These results again indicate that the proposed method was able to well outperform the expected precision of random sampling given that the target behaviour comprises only 23.20% of the dataset.

In addition to our investigation of the effectiveness of the proposed method, we also used the video captured by our bio-loggers to analyse the relationship between ARS behaviour and group formation in streaked shearwaters[20]. A count was done of the number of birds visible in each frame of video captured, with each frame also labelled with the animal's ID and its current behaviour, i.e., ARS vs. non-ARS (transit or stationary). Since the counts came from consecutive video frames, we chose to analyse this data using a generalised linear mixed model (GLMM) with Gaussian error distribution, with the individual factors (based on the animal collecting the video) treated as random effects. Using GLMM analysis, we found a significant difference in the number of birds visible during ARS behaviour ($t = -41.919$; $df = 4.820 \times 10^4$; $p = 2 \times 10^{-16}$ (two-sided), effect size ($r^2$) = 0.070). The number of data points for ARS was 29,195 and for non-ARS was 23,741 (see Supplementary Fig. 4 for a chart of the dataset distribution). These results support the hypothesis that streaked shearwaters forage in flocks during ARS.

## Discussion

While several previous studies have proposed energy-efficient machine learning algorithms for use with larger wearable devices[21–23], few have explored space-efficient models for use on devices with extreme memory constraints, such as an ATmega328P MCU[24–26]. Even in these latter studies, the assumed memory constraints were based on the total RAM available on the devices (i.e., 2 KB), with two studies mostly generating models larger than 2 KB in size[25,26] and only one attempting to optimise its model sizes to below 1 KB[24]. Furthermore, the models used in that final study were based on precomputed feature vectors[24], meaning that the space needed for feature extraction from raw time-series data was not considered when calculating their memory use.

In addition, several previous studies involving bio-loggers have introduced trigger mechanisms that can be used to control when high-cost sensors are activated based on coarse-level characterisations of behaviour, e.g., underwater vs. surface activity, with many of these studies focusing on controlling animal-borne cameras such as the one used in this study[10,11,13,27–32]. In contrast, our method can be used to distinguish between complex behaviours at a finer scale, allowing us to target a specific behaviour; thereby greatly increasing the likelihood that interesting behaviours will be captured. A case in point is the video captured by AIoA of intraspecific kleptoparasitism at sea by a black-tailed gull (Fig. 1e, Supplementary Movie 1). In addition, three of the foraging videos captured using AIoA included footage of the gulls foraging for flying insects over the sea (Supplementary Movie 3, Fig. 2e, f), a previously unreported behaviour. Until now, insects

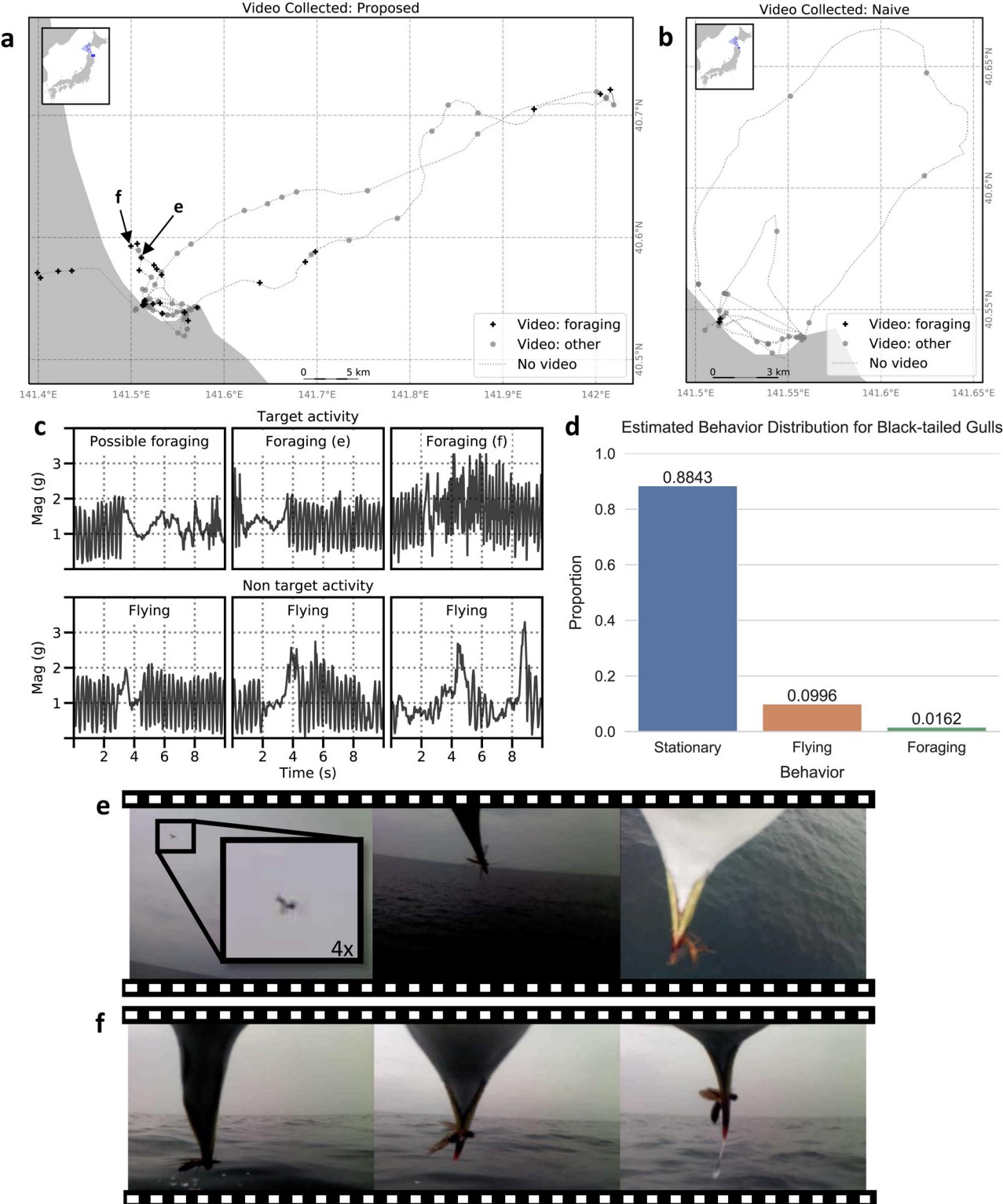

found in traditionally used stomach-content analyses have been considered to be preyed upon over land[18]. As this example shows, by focusing the bio-loggers' data collection on a target behaviour, we can increase the probability with which new findings related to that behaviour are discovered.

While the need for intelligent methods for supporting data collection in the wild has motivated a wide range of previous studies[1,2,33], this is the first study to our knowledge to use machine learning on board animal-borne data loggers to support

data collection in the wild. Using machine learning, we can focus data collection by high-cost sensors on interesting but infrequent behaviours (e.g., 1.6% occurrence rate), greatly reducing the number of bio-loggers required to collect the same amount of data from interesting behaviours when compared to naive data collection methods. Furthermore, since reducing or limiting the weight of data loggers is an important aspect of experimental design[34], our approach can be used by researchers to reduce battery requirements in order to either reduce device mass or

**Fig. 2 Results of AI video control for black-tailed gulls. a** GPS tracks marked with the locations of videos collected by bio-loggers using the proposed method. The letters e and f indicate the locations where the video frames shown in (**e**) and (**f**) were collected. **b** GPS tracks marked with the locations of videos collected by bio-loggers using the naive method (periodic sampling). **c** Examples of acceleration data (shown as magnitude of acceleration) collected around the time of video camera activation on bio-loggers using the proposed method. Cells *Foraging* (**e**) and *Foraging* (**f**) show the acceleration data that triggered the camera to record the video frames shown in (**e**) and (**f**). Note that the camera is activated based on a 1-s window of data, which corresponds to a window extracted from around the 2- to 4-s mark for each example. As shown in these charts, while acceleration data can be used to detect the target behaviour, it is difficult to avoid false positives due to the similarity between the target behaviour and other anomalous movements in the sensor data. **d** Estimated distribution of behaviours based on the 116 h of acceleration data collected. **e** Frames taken from video captured using AIoA of a black-tailed gull catching an insect in mid-air while flying over the ocean. **f** Frames taken from video captured using AI on Animals (AIoA) of a black-tailed gull plucking an insect off the ocean surface. Supplementary Data 1 provides source data of this figure.

---

increase the amount of data collected using a given device mass. We anticipate this work will provide motivation for more wide-spread research into AIoA, further improving its ability to control resource-intensive sensors such as video cameras, microphones and EEGs. Furthermore, AIoA could even be applied to other applications such as controlling what data is transmitted from devices over the low-bandwidth connections used with satellite relay tags[35] and detecting the poaching of endangered species with real-time anti-poaching tags[36].

## Methods

**Video bio-logger hardware.** Supplementary Fig. 1a shows an example of the video bio-loggers used during this study. It measures 85 mm length × 35 mm width × 15 mm height. The bio-loggers were attached to either the bird's back or abdomen by taping them to the bird's feathers using waterproof tape. When attaching the bio-logger to a bird's abdomen, a harness made of Teflon ribbon (Bally Ribbon Mills, USA) was also used. When working with streaked shearwaters, the bio-loggers used a 3.7 V 600 mAh battery and weighed approximately 26–27 g. When working with black-tailed gulls, the bio-loggers used a 3.7 V 720 mAh battery and weighed approximately 30 g.

The bio-loggers are controlled by an ATmega328 MCU (32 KB programme memory, 2 KB RAM) and have an integrated video camera (640 × 480, 15 FPS) that can be controlled by the MCU, with the video data streamed directly to its own dedicated storage. Note that digital cameras such as the one used in this bio-logger have a delay of several seconds from powering on to when they can begin recording, which in the case of our bio-logger resulted in a 2- to 3-s delay between when the MCU signals the start of recording and the actual start of recording when attempting to save energy by powering off the camera when not in use (see also Yoshino et al.[13] for another example of this camera delay). Our bio-loggers also include several low-cost sensors that are controlled by the MCU (see Supplementary Fig. 1b). Each of these sensors can be used by the MCU as input for AIoA applications (e.g., camera control) and can be archived to long-term storage for analysis upon device retrieval. The bio-loggers had an average battery life of approximately 2 h when continuously recording video and approximately 20 h when recording from all other (i.e., only low-cost) sensors.

**Activity recognition method.** We achieve AIoA by employing machine learning to conduct activity (behaviour) recognition on board the bio-logging devices. We do this by training an activity recognition model in advance using low-cost sensor data that has been labelled by an ecologist to identify the behaviours that he/she wants to capture. In the case of the black-tailed gulls, we use accelerometer-based features since they can be used to detect the body movements of the animals with only a small (e.g., 1-s) delay between when data collection begins and when behaviours can first be detected. Such features are often used when detecting body movements in human activity recognition studies in order to recognise activities such as running and eating[37]. For animal-based AI, such body movements can be useful to detect similar types of behaviours, such as flying and foraging[38]. See Fig. 3 for an example of how such accelerometer-based features can be extracted from raw data and used to train a decision tree classifier model. The features were extracted from 1-s windows of 25 Hz acceleration data, with the raw 3-axis acceleration data converted to net magnitude of acceleration data prior to feature extraction. The activity recognition processes were run once per second on the 1-s windows of data, allowing us to detect target behaviours within about 1 s of their start. See Supplementary Table 1 for descriptions and estimated sizes for all the features used in this study. In addition, Supplementary Fig. 5a shows the acceleration-based features ranked by their Normalised Gini Importance when used to classify behaviours for black-tailed gulls.

The energy-saving microcontroller units (MCUs) in bio-loggers tend to have limited memory and low computing capability, which makes it difficult to run the computationally expensive processes needed for the pretrained machine learning models on board the bio-loggers. Therefore, this study proposes a method for generating space-efficient, i.e., programme memory efficient, decision tree classifier

models that can be run on such MCUs. Decision trees are well suited for use on MCUs, since the tree itself can be implemented as a simple hierarchy of conditional logic statements, with most of the space needed for the tree being used by the algorithms needed to extract meaningful features from the sensor data, such as the *variance* or *kurtosis* of 1-s windows of data. In addition, since each data segment is classified by following a single path through the tree from the root node to the leaf node that represents that data segment's estimated class, an added benefit of using a tree structure is that the MCU only needs to extract features as they are encountered in the path taken through the tree, allowing the MCU to run only a subset of the feature extraction processes for each data segment. However, the feature extraction algorithms needed by the tree can be prohibitively large, e.g., *kurtosis* requires 680 bytes (Supplementary Table 1), when implemented on MCUs that typically have memory capacities measured in kilobytes, e.g., 32 KB, which is already largely occupied by the functions needed to log sensor data to storage.

Standard decision tree algorithms, e.g., scikit-learn's default algorithm, build decision trees that maximise classification accuracy with no option to weight the features used in the tree based on a secondary factor such as memory usage[39]. The trees are built starting from the root node, with each node in the tree choosing from among all features the one feature that can best split the training data passed to it into subsets that allow it to differentiate well between the different target classes. A new child node is then created for each of the subsets of training data output from that node, with this process repeating recursively until certain stopping conditions are met, e.g., the subsets generated by a node reach a minimum size. Figure 4b shows an example of a decision tree built using scikit-learn's default decision tree classifier algorithm using the black-tailed gull data, which results in an estimated memory footprint of 1958 bytes (Supplementary Table 1). Note that since the basic system functions needed to record sensor data to long-term storage already occupy as much as 95% of the bio-logger's flash memory, incorporating a decision tree with this large of a memory footprint can cause the programme to exceed the bio-logger's memory capacity (see the bio-logger source code distributed as Supplementary Software for more details).

In this study, we propose a method for generating decision tree classifiers that can fit in bio-loggers with limited programme memory (e.g., 32 KB) that is based on the random forests algorithm[40], which is a decision tree algorithm that injects randomness into the trees it generates by restricting the features compared when creating the split at each node in a tree to a randomly selected subset of the features, as opposed to the standard decision tree algorithm that compares all possible features at each node, as was described above. Our method modifies the original random forests algorithm by using *weighted* random selection when choosing the subset of features to compare when creating each node. Figure 4a illustrates the weighted random selection process used by our method. We start by assigning each feature a weight that is proportional to the inverse of its size. We then use these weights to perform weighted random selection when selecting a group of features to consider each time we create a new node in the tree, with the feature used at that node being the best candidate from amongst these randomly selected features.

Using our method for weighted random selection of nodes described above, we are then able to generate randomised trees that tend to use less costly features. When generating these trees, we can estimate the size of each tree based on the sizes of the features used in the tree and limit the overall size by setting a threshold and discarding trees above that threshold. Figure 4c shows an example batch of trees output by our method where we have set a threshold size of 1000 bytes. We can then select a single tree from amongst these trees that gives our desired balance of cost to accuracy. In this example, we have selected the tree shown in Fig. 4d based on its high estimated accuracy. Comparing this tree to Fig. 4b, we can see that our method generated a tree that is 42% the size of the default tree while maintaining close to the same estimated accuracy. We developed our method based on scikit-learn's (v.0.20.0) RandomForestClassifier.

In addition, in order to achieve robust activity recognition, our method also has the following features: (i) robustness to sensor positioning, (ii) robustness to noise, and (iii) reduction of sporadic false positives. Note that robustness to noise and positioning are extremely important when deploying machine learning models on bio-loggers, as the models will likely be generated using data collected in previous years, possibly using different hardware and methods of attachment. While there is a potential to improve prediction accuracy by removing some of these variables,

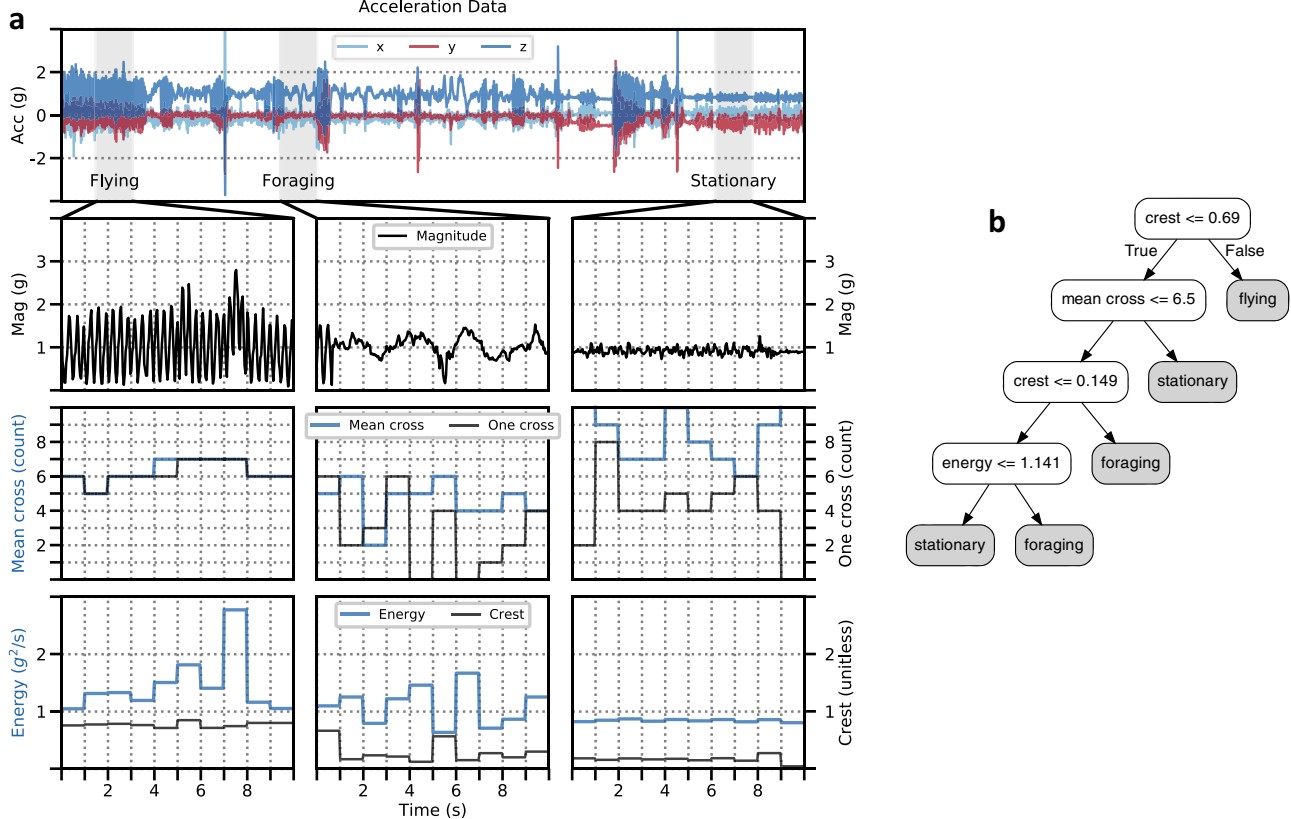

**Fig. 3 Generating decision trees from acceleration data. a** We start by converting the raw three-axis data (row one) into magnitude of acceleration values (row two) and segmenting the data into 1-s windows. We then extract the *ACC* features listed in Supplementary Table 1 from each window. Rows three and four show examples of the features extracted, which can be used to differentiate between the behaviours. For example, *Crest* can be used to identify *Flying* behaviour, since its values are higher for *Flying* than for the other two behaviours. **b** An example decision tree generated from the feature values shown in the lower two rows of (**a**), with each leaf (grey) node representing a final predicted class for a 1-s segment of data. Supplementary Data 1 provides source data of this figure.

e.g., by collecting from the same animal multiple times using the same hardware, moving to more animal-dependent models is generally not practical as care must be taken to minimise the handling of each animal along with the amount of time the animals spend with data loggers attached[34]. See "Robust activity recognition" for more details.

**GPS features**. Due to the low resolution of GPS data (e.g., metre-level accuracy), GPS-based features cannot detect body movements with the same precision as acceleration-based features, but are useful when analysing patterns in changes in an animal's location as it traverses its environment. For animal-based AI, these features can be used to differentiate between different large-scale movement patterns, such as transit versus ARS. In this study, we used GPS-based features to detect ARS by streaked shearwaters. These features were extracted once per minute from 1/60th Hz GPS data using 10-min windows. Supplementary Fig. 5b shows these features ranked by their Normalised Gini Importance when used to classify behaviours for streaked shearwaters. Supplementary Fig. 2 shows an example of two such 10-min windows that correspond to target (ARS) and non-target (transit) behaviour, along with several examples of GPS-based features extracted from those windows. Supplementary Table 1 describes all the GPS features used in this study along with their estimated sizes when implemented on board our bio-logger. Note that the *variance* and *mean cross* features were extracted after first rotating the GPS positions around the mean latitude and longitude values at angles of 22.5°, 45.0°, 67.5°, and 90.0° in order to find the orientation that maximised the variance in the longitude values (see Supplementary Table 1, feature Y: *rotation*). This was done to provide some measure of rotational invariance to these values without the need for a costly procedure such as principal component analysis. The primary and secondary qualifiers for these features refer to whether the feature was computed on the axis with maximised variance vs. the perpendicular axis, respectively.

**Robust activity recognition**. In this study, we also incorporated two methods for improving the robustness of the recognition processes in the field. First, we addressed how loggers can be attached to animals at different positions and orientations, such as on the back to maximise GPS reception or on the abdomen to improve the camera's field of view during foraging. For example, during our case

study involving black-tailed gulls, the AI models were trained using data collected from loggers mounted on the birds' backs, but in many cases were used to detect target behaviour on board loggers mounted on the birds' abdomens. We achieved this robustness to positioning by converting the three-axis accelerometer data to net magnitude of acceleration values, thereby removing the orientation information from the data. To test the robustness of the magnitude data, we evaluated the difference in classification accuracy between raw three-axis acceleration data and magnitude of acceleration data when artificial rotations of the collection device were introduced into the test data. In addition, we also evaluated the effectiveness of augmenting the raw three-axis data with artificially rotated data as an alternative to using the magnitude of acceleration data. These results are shown in Supplementary Fig. 6a. Note that the results for the magnitude of acceleration data are constant across all levels of test data rotation, since the magnitudes are unaffected by the rotations. Based on these results, we can see that extracting features from magnitude of acceleration data allows us to create features that are robust to the rotations of the device that can result from differences in how the device is attached to an animal.

Next, we addressed the varying amount of noise that can be introduced into the sensor data stemming from how loggers are often loosely attached to a bird's feathers. We achieved this noise robustness by augmenting our training dataset with copies of the dataset that were altered with varying levels of random artificial noise, with this noise added by multiplying all magnitude of acceleration values in each window of data by a random factor. We tested the effect of this augmentation by varying the amount of artificial noise added to our training and testing data and observing how the noise levels affected performance (see Supplementary Fig. 6b). Based on these results, the training data used for fieldwork was augmented using the 0.2 level. Note that at higher levels of simulated noise (test noise greater than 0.15) the training noise settings of 0.25 and 0.3 both appear to outperform the 0.2 setting. However, since these results are based solely on laboratory simulations, we chose to use the more conservative setting of 0.2 in the field.

**Reduction of sporadic false positives**. When activating the camera to capture target behaviour, it is possible to reduce the number of false positives (i.e., increase our confidence in the classifier's output) by considering multiple consecutive outputs from the classifier before camera activation. We accomplish this using two

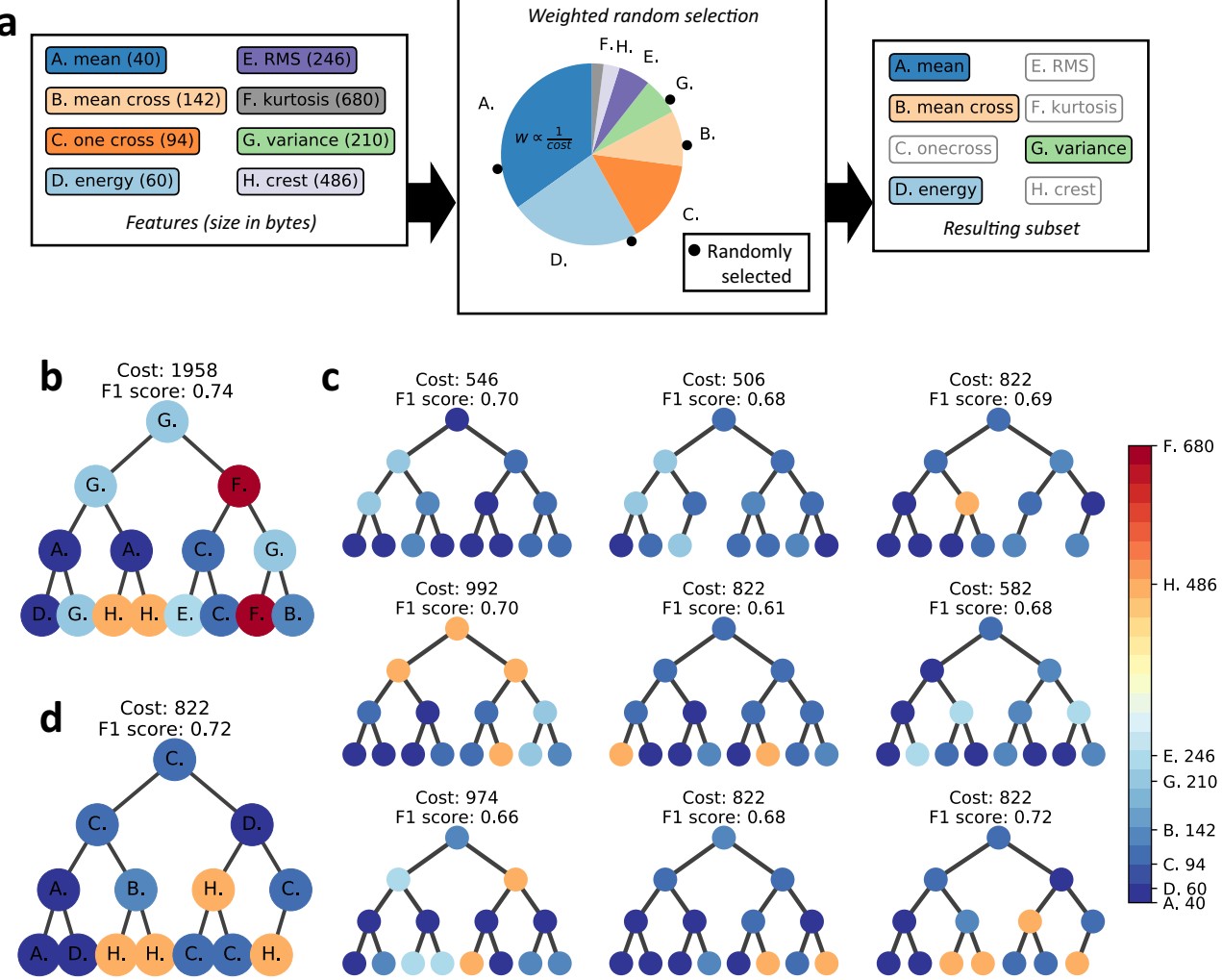

**Fig. 4 Generating space-efficient trees. a** Our process for the weighted random selection of features. We start with a list of features along with their required programme memory sizes in bytes (first panel). Each feature is assigned a weight proportional to the inverse of its size, illustrated using a pie chart where each feature has been assigned a slice proportional to its weight (second panel). We then perform weighted random selection to choose the subset of features that will be used when creating a new node in the tree. In this example, we have randomly placed four dots along the circumference of the circle to simulate the selection of four features (second panel). The resulting subset of features will then be compared when making the next node in the decision tree (third panel). **b** Example decision tree built using scikit-learn's default decision tree classifier algorithm using the black-tailed gull data described in "Methods". Each node is coloured based on its corresponding feature's estimated size in bytes when implemented on board the bio-logger (scale shown in the colour bar). **c** Several space-efficient decision trees generated using the proposed method from the same data used to create the tree in (**b**). **d** Example space-efficient tree selected from the trees shown in (**c**) that costs much less than the default tree in (**b**) while maintaining almost the same accuracy.

methods. In the first, we assume that the classifier can reliably detect the target behaviour throughout its duration, allowing us to increase our confidence in the classifier's output by requiring multiple consecutive detections of the target behaviour before activating the camera. We employed this method when detecting ARS behaviour for streaked shearwaters using GPS data, since the characteristics of the GPS data that allow for detection of the target behaviour were expected to be consistent throughout its duration, with the number of consecutive detections required set to 2.

In the second method, we assume that the classifier cannot reliably detect the target behaviour throughout its duration, since the actions corresponding to the target behaviour that the classifier can reliably detect occur only sporadically throughout its duration. In this case, we can instead consider which behaviours were detected immediately prior to the target behaviour. When controlling the camera for black-tailed gulls, we assume that detection of the target behaviour (foraging) is more likely to be a true positive after detecting flying behaviour, since the birds typically fly when searching for their prey. Therefore, we required that the logger first detect five consecutive windows of flying behaviour to enter a flying state in which it would activate the camera immediately upon detecting foraging. This flying state would time out after ten consecutive windows of stationary behaviour. Note that in this case, while the intervals of *detectable* target behaviour

may be short and sporadic, the overall duration of the target behaviour is still long enough that we can capture video of the behaviour despite the delay between behaviour detection and camera activation (see "Video bio-logger hardware" for details).

**Procedures of experiment of black-tailed gulls**. We evaluated the effectiveness of our method by using AIoA-based camera control on board ten bio-loggers that were attached to black-tailed gulls (on either the bird's abdomen or back) from a colony located on Kabushima Island near Hachinohe City, Japan[18], with the AI trained to detect possible foraging behaviour based on acceleration data. The possible foraging events were identified based on dips in the acceleration data. The training data used for the AI was collected at the same colony in 2017 using Axy-Trek bio-loggers (TechnoSmArt, Roma, Italy). These Axy-Trek bio-loggers were mounted on the animals' backs when collecting data. Along with the AIoA-based bio-loggers, three bio-loggers were deployed using a naive method (periodic sampling), with the cameras controlled by simply activating them once every 15 min. All 13 loggers recorded 1-min duration videos.

Sample size was determined by the time available for deployment and the availability of sensor data loggers. The birds were captured alive at their nests by

hand prior to logger deployment and subsequent release. Loggers were fitted externally within 10 min to minimise disturbance. Logger deployment was undertaken by the ecologists participating in this study. Loggers that suffered hardware failures (e.g., due to the failure of the waterproofing material used on some loggers) were excluded.

**Ethics statement**. All experimental procedures were approved by the Animal Experimental Committee of Nagoya University. *Black-tailed gulls*: the procedures were approved by the Hachinohe city mayor, and the Aomori Prefectural Government. *Streaked shearwaters*: the study was conducted with permits from the Ministry of the Environment, Japan.

**Statistics and reproducibility**. Fisher's exact tests were done using the exact2x2 package (v. 1.6.3) of R (v. 3.4.3). The GLMM analysis was conducted using the lmerTest package (v. 2.0–36) of R (v. 3.4.3). In regards to reproducibility, no experiments as such were conducted, rather our data are based on tracked movements of individual birds.

**Reporting summary**. Further information on research design is available in the Nature Research Reporting Summary linked to this article.

## Data availability
The data from this study are available from the corresponding author upon reasonable request.

## Code availability
Our code run on the bio-loggers is written in C++. Code needed to replicate our findings and hardware diagrams of the bio-loggers used in this study can be found at: http://www-mmde.ist.osaka-u.ac.jp/~maekawa/paper/supple/logger/CodeAvailability.zip and https://doi.org/10.5281/zenodo.4007788[41].

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

## Acknowledgements
We thank Rory P. Wilson, Flavio Quintana, Agustina Gómez Laich, Takashi Yamamoto, Yasue Kishino, and Kazuya Murao for suggestions and comments on this work. This study is partially supported by JSPS KAKENHI JP16H06539, JP16H06541 and JP16H06536.

## Author contributions

J.K. performed the method design, software implementation, data collection, data analysis and paper writing. H.S., S.M. and Y.M. performed the data collection and data analysis. M.S. performed the software implementation and data collection. T.M. conceived and directed the study, and performed method design, data collection, data analysis and paper writing. J.N. performed the hardware design. K.Y. performed data collection, data analysis and paper writing.

## Competing interests

The authors declare no competing interests.
