## [Peer Review File · Communications Biology]

Reviewers' comments:

Reviewer #1 (Remarks to the Author):

This study presents a new method that uses signals from low cost sensors such as accelerometers, to activate high cost sensors, such as video cameras. This results in memory savings, allows the combined units to log for longer, and reduces post-processing requirements. The manuscript is well written, the machine learning approach is clear and builds on approaches that are now reasonably widespread in the analysis of tagging data – although the application of such techniques onboard the logger itself is novel. One of the strengths is the use of two different data types in the study systems as proxies for possible foraging behaviour. Overall, the approach will no doubt be useful for the bio-logging community, but the extent of the advance is not commensurate with that required for Nature Communications and will be difficult for others to implement given the reliance on commercial tag providers.

Nonetheless, there are good arguments for advancing this approach, as it would lead to (i) a reduction in overall device mass (and therefore tag effects) and/ or (ii) an increase in data for a given device mass. I miss a wider discussion of the systems where either of these outcomes might be most valuable.

Finally, the value of the approach presented here depends on the performance of behavioural recognition algorithms, yet the track record of machine learning techniques in identifying less common behaviours is not great. Indeed, the precision and recall presented here are also not brilliant, although the context is rather different as here they need to be robust to individual differences and noise introduced by a variable environment. It would be useful to address the conflicting needs of generality and accuracy in the manuscript.

P3 lines 35-7. Many users now only record bursts of data to get around battery and memory restrictions. I suggest making this clearer.

P4, last paragraph. Perhaps I'm missing something, but I find this apparent trial of the precision of the 2 techniques rather misleading, as periodic sampling regimes are not set up to capture all instances of a given behaviour.

P5, the finding that gulls forage for insects over the sea is interesting, but it does not represent a major ecological insight.

Reviewer #2 (Remarks to the Author):

A central challenge of biologging is to maximize the collection of relevant data while working within the constraints of battery life and device processing and storage capacity, all of which are limited by the body weight of the tagged animal. This manuscript introduces a method of using artificial intelligence to tailor the data collection activities of bio-logging devices to detect and capture behaviors of interest, thereby significantly increasing the amount of relevant data yielded by a given device. In developing this approach, the authors have creatively solved a number of challenges, including:

- Training data collection algorithms on data collected using different methodologies – this is important as it is not always practical or ethical to conduct pilot tagging of animals just to collect training data. If existing but imperfectly-matched data can be used for training, this makes the AIOA method more readily deployable, and, as the authors point out, speaks to the method's robustness.
- Reducing the computational capacity required to generate reliable behavior classifications by modifying standard decision tree models using weighted randomized feature selection. This makes

it possible for low-memory devices to perform the analyses necessary to reliably identify desirable windows for data recording.

Key results include a significant increase in the amount of relevant data collected (compared to a naïve, periodic sampling approach), and the documentation of a novel foraging behavior (insect capture over water). The Discussion does a nice job of highlighting the advances of the described method over previous approaches and proposing other potential applications.

Overall, I find this to be a very novel and timely manuscript that will be of broad interest within the bio-logging, conservation, and behavioral ecology communities. Use of AI on bio-logging devices is an important direction for optimizing the effectiveness of these devices, and this paper represents a significant step in this direction. Although some of the computational processes are complex and technical, I found the authors' explanations and supporting figures (particularly Fig 3 and 4) to be very clear, both in the main text and the supplement. The included videos do a good job of illustrating the usefulness of the behavioral data yielded by the described method. I find no major flaws with the manuscript, but propose some minor changes below.

Line-by-line comments

- L 64-73: This paragraph seems more like Methods content than Results
- Figure 2: To me the frames in parts E and F seem a bit disconnected from the rest of the figure, and superfluous given that the actual videos are included as supplementary files. I suggest either incorporating these frames with the rest of the figure (perhaps by highlighting the point of video capture in panel A, and including the 'triggering' acceleration data for these clips in panel C) or removing E and F from the figure.
- L90-97: Most of this material seems better-suited to the Discussion section as it is explaining the significance of the result rather than the result itself.
- L108-109: I think it would be clearer to say 'the proposed method captured ARS in 19 of the 20 recorded videos'

Reviewer #3 (Remarks to the Author):

This paper presents a new method for controlling on-board data collection devices in animal tracking studies. The authors use a small decision tree algorithm to trigger an on-board camera whenever the algorithm expects the animal to be foraging. They demonstrate the result is a more efficient data collection process where video is collected only at key points in the activities of the animal.

This is a nice and interesting paper. It is well-written, clear and provides a solid evaluation of the method. My only concern is that this is not sufficiently general interest for this journal. This is not a method that is broadly applicable across study systems and the memory constraints of the on-board devices mean the algorithm will be limited to a small number of use cases. In my view this paper would be better suited to a specialised journal.

My only other criticism of the paper was the use of AI to refer to their method in the title and throughout. While AI is a nebulous term I don't think the algorithm defined in Fig3B can be considered AI.

We would like to take this opportunity to thank all the reviewers for their constructive comments on our original submission, which have helped us in improving the quality of the paper. We would also like to thank you for all your help throughout this process.

Reviewer's condition	Our response
R1.1) This study presents a new method that uses signals from low cost sensors such as accelerometers, to activate high cost sensors, such as video cameras. This results in memory savings, allows the combined units to log for longer, and reduces post-processing requirements. The manuscript is well written, the machine learning approach is clear and builds on approaches that are now reasonably widespread in the analysis of tagging data – although the application of such techniques onboard the logger itself is novel. One of the strengths is the use of two different data types in the study systems as proxies for possible foraging behaviour. Overall, the approach will no doubt be useful for the bio-logging community, but the extent of the advance is not commensurate with that required for Nature Communications and will be difficult for others to implement given the reliance on commercial tag providers. Nonetheless, there are good arguments for advancing this approach, as it would lead to (i) a reduction in overall device	We have updated the paper to include a discussion on how our approach could lead to either of these outcomes. Lines 131 – 134. Furthermore, since reducing or limiting the weight of data loggers is an important aspect of experimental design³³, our approach can be used by researchers to reduce battery requirements in order to either reduce device mass or increase the amount of data collected using a given device mass.

mass (and therefore tag effects) and/ or (ii) an increase in data for a given device mass. I miss a wider discussion of the systems where either of these outcomes might be most valuable.	
R1.2) Finally, the value of the approach presented here depends on the performance of behavioural recognition algorithms, yet the track record of machine learning techniques in identifying less common behaviours is not great. Indeed, the precision and recall presented here are also not brilliant, although the context is rather different as here they need to be robust to individual differences and noise introduced by a variable environment. It would be useful to address the conflicting needs of generality and accuracy in the manuscript.	We have added a short discussion of why animal- and environment-independent models are important when deploying machine learning models on data loggers. Lines 220 – 227. Note that robustness to noise and positioning are extremely important when deploying machine learning models on bio-loggers, as the models will likely be generated using data collected in previous years, possibly using different hardware and methods of attachment. While there is a potential to improve prediction accuracy by removing some of these variables, e.g., by collecting from the same animal multiple times using the same hardware, moving to more animal-dependent models is generally not practical as care must be taken to minimize the handling of each animal along with the amount of time the animals spend with data loggers attached³³.
R1.3) P3 lines 35-7. Many users now only record bursts of data to get around battery and memory restrictions. I suggest making this clearer.	We have updated these lines based on your feedback. Lines 34 – 38. Although there have been extraordinary improvements in the sensors and storage capacities of bio-loggers since the first logger was attached to a Weddell seal⁴⁻⁹, their data collection strategies have remained relatively simple: record data continuously, record data in bursts (e.g., periodic sampling), or use manually determined thresholds to detect basic collection criteria such as a minimum depth, acceleration threshold, or illumination level¹⁰⁻¹⁷.
R1.4) P4, last paragraph. Perhaps I'm missing something, but I find this apparent trial of the precision of the 2 techniques rather misleading, as periodic sampling regimes are not set up to capture all	We feel that periodic sampling is an appropriate baseline with which to compare our method, as it is commonly used to deal with battery limitations when collecting video data using bio-loggers. Additionally, machine learning studies often use randomized methods that assume no prior knowledge for comparison with their proposed methods, with periodic sampling filling that role in our study. We have added multiple references that show examples of periodic sampling that also serve as examples of burst recording. See our response to R1.3 for further details.

instances of a given behaviour.	
R1.5) P5, the finding that gulls forage for insects over the sea is interesting, but it does not represent a major ecological insight.	We have reworded these lines in order to tone down the implied significance of this finding. Lines 118 – 123. Additionally, three of the foraging videos captured using AloA included footage of the gulls foraging for flying insects over the sea (Supplementary Movie 3, Fig. 2e and f), a previously unreported behaviour. Until now, insects found in traditionally used stomach-content analyses have been considered to have been preyed upon over land¹⁸. As this example shows, by focusing the bio-loggers' data collection on a target behaviour, we can increase the probability with which new findings related to that behaviour are discovered.
R2.1) A central challenge of biologging is to maximize the collection of relevant data while working within the constraints of battery life and device processing and storage capacity, all of which are limited by the body weight of the tagged animal. This manuscript introduces a method of using artificial intelligence to tailor the data collection activities of bio-logging devices to detect and capture behaviors of interest, thereby significantly increasing the amount of relevant data yielded by a given device. In developing this approach, the authors have creatively solved a number of challenges, including: - Training data collection algorithms on data collected using different methodologies – this is important as it is not always practical or ethical to conduct pilot tagging of animals just to collect training data. If existing but imperfectly-matched data	We have moved this paragraph to the Methods section and have modified the beginning of the following paragraph to add a short introduction on how the results were obtained. Lines 66 – 69. We evaluated the effectiveness of our method by using AloA-based camera control on board 10 bio-loggers that were attached to black-tailed gulls from a colony located on Kabushima Island near Hachinohe City, Japan¹⁸, with the AI trained to detect possible foraging behaviour based on acceleration data. Lines 229 – 238. We evaluated the effectiveness of our method by using AloA-based camera control on board 10 bio-loggers that were attached to black-tailed gulls (on either the bird's abdomen or back) from a colony located on Kabushima Island near Hachinohe City, Japan¹⁸, with the AI trained to detect possible foraging behaviour based on acceleration data. The possible foraging events were identified based on dips in the acceleration data. The training data used for the AI was collected at the same colony in 2017 using Axy-Trek bio-loggers (TechnoSmArt, Roma, Italy). These Axy-Trek bio-loggers were mounted on the animals' backs when collecting data. Along with the AloA-based bio-loggers, three bio-loggers were deployed using a naive method (periodic sampling), with the cameras controlled by simply activating them once every 15 minutes. All 13 loggers recorded 1-minute duration videos.

can be used for training, this makes the AloA method more readily deployable, and, as the authors point out, speaks to the method's robustness.

- Reducing the computational capacity required to generate reliable behavior classifications by modifying standard decision tree models using weighted randomized feature selection. This makes it possible for low-memory devices to perform the analyses necessary to reliably identify desirable windows for data recording.

Key results include a significant increase in the amount of relevant data collected (compared to a naïve, periodic sampling approach), and the documentation of a novel foraging behavior (insect capture over water). The Discussion does a nice job of highlighting the advances of the described method over previous approaches and proposing other potential applications.

Overall, I find this to be a very novel and timely manuscript that will be of broad interest within the bio-logging, conservation, and behavioral ecology communities. Use of AI on bio-logging devices is an important direction for optimizing the effectiveness of these devices, and this paper represents a significant

step in this direction. Although some of the computational processes are complex and technical, I found the authors' explanations and supporting figures (particularly Fig 3 and 4) to be very clear, both in the main text and the supplement. The included videos do a good job of illustrating the usefulness of the behavioral data yielded by the described method. I find no major flaws with the manuscript, but propose some minor changes below. Line-by-line comments L 64-73: This paragraph seems more like Methods content than Results	
R2.2) Figure 2: To me the frames in parts E and F seem a bit disconnected from the rest of the figure, and superfluous given that the actual videos are included as supplementary files. I suggest either incorporating these frames with the rest of the figure (perhaps by highlighting the point of video capture in panel A, and including the 'triggering' acceleration data for these clips in panel C) or removing E and F from the figure.	We have updated Figure 2 to better incorporate subfigures (e) and (f). Subfigure (a) was updated to indicate the locations where (e) and (f) were captured, while subfigure (c) was updated to include the data segment that triggered the video corresponding to (f) and was updated to indicate which data segment corresponds to each of the videos. The figure legend was also updated to explain the connections between subfigures (a), (c), (e), and (f). An updated version of Figure 2 along with its figure legend are included below.
R2.3) L90-97: Most of this material seems better-suited to the Discussion section as it is explaining the significance of the result rather than the result itself.	We have moved these lines to the Discussion section. Lines 118 – 123. Additionally, three of the foraging videos captured using AIOA included footage of the gulls foraging for flying insects over the sea (Supplementary Movie 3, Fig. 2e and f), a previously unreported behaviour. Until now, insects found in traditionally used stomach-content analyses have been considered to have been preyed upon over land¹⁸. As this example shows, by focusing the bio-loggers' data collection on a target behaviour,

	we can increase the probability with which new findings related to that behaviour are discovered.
R2.4) L108-109: I think it would be clearer to say ‘the proposed method captured ARS in 19 of the 20 recorded videos’	We have modified those lines to more clearly state what we meant. Lines 94 – 97. Furthermore, of the 20 videos of ARS that were captured by the proposed method and the baseline method, the proposed method captured 19 of those 20 videos, thereby playing a key role in providing video evidence in support of the hypothesis that streaked shearwaters forage in groups during ARS.
R3.1) This paper presents a new method for controlling on-board data collection devices in animal tracking studies. The authors use a small decision tree algorithm to trigger an on-board camera whenever the algorithm expects the animal to be foraging. They demonstrate the result is a more efficient data collection process where video is collected only at key points in the activities of the animal. This is a nice and interesting paper. It is well-written, clear and provides a solid evaluation of the method. My only concern is that this is not sufficiently general interest for this journal. This is not a method that is broadly applicable across study systems and the memory constraints of the on-board devices mean the algorithm will be limited to a small number of use cases. In my view this paper would be better suited to a specialised journal.	We believe that our method will generate broad interest and that it has a wide range of use cases on data logging systems. As was mentioned by Reviewer #2, maximizing relevant data collection in devices with strict constraints on battery life and device processing capability is a fundamental challenge in the bio-logging community. Similarly, in condition R.1.1, Reviewer #1 also highlighted the fact that our technique potentially reduces overall device mass and increases the data collected using a given mass.
R3.2) My only other criticism of the paper was	We have modified the title and text to also use the term “machine learning” in order to clarify how our method relates to

the use of AI to refer to their method in the title and throughout. While AI is a nebulous term I don't think the algorithm defined in Fig3B can be considered AI.	the AI field. However, we feel that it is good to also use the term AI, since machine learning is a type of AI and since this term will draw more interest from potential readers, which also relates to R3.1. Title. AI on animals: transforming animal tracking systems with machine learning Lines 62 – 63. See Methods for a description of the machine learning algorithm used when recognizing behaviours on board the bio-loggers. Lines 84 – 85. Along with the above evaluation that used acceleration data to train the machine learning models, we also evaluated the proposed method when training the models with GPS data. Lines 125 – 131. While the need for intelligent methods for supporting data collection in the wild has motivated a wide range of previous studies^{1,2,32}, this is the first study to our knowledge to use machine learning on board animal-borne data loggers to support data collection in the wild. Using machine learning, we can focus data collection by high-cost sensors on interesting but infrequent behaviours (e.g., 1.6% occurrence rate), greatly reducing the number of bio-loggers required to collect the same amount of data from interesting behaviours when compared to naive data collection methods. Lines 161 – 163. The energy-saving microcontroller units (MCUs) in bio-loggers tend to have limited memory and low computing capability, which makes it difficult to run the computationally expensive processes needed for the pretrained machine learning models on board the bio-loggers.
---	--

Fig. 2 Results of AI video control for black-tailed gulls. **a** GPS tracks marked with the locations of videos collected by bio-loggers using the proposed method. The letters *e* and *f* indicate the locations where the video frames shown in **(e)** and **(f)** were collected. **b** GPS tracks marked with the locations of videos collected by bio-loggers using the naive method (periodic sampling). **c** Examples of acceleration data (shown as magnitude of acceleration) collected around the time of video camera activation on bio-loggers using the proposed method. Cells *Foraging (e)* and *Foraging (f)* show the acceleration data that triggered the camera to record the video frames shown in **(e)** and **(f)**. Note that the camera is activated based on a 1-second window of data, which corresponds to a window extracted from around the 2- to 4-second mark for each example. As shown in these charts, while acceleration data can be used to detect the target behaviour, it is difficult to avoid false positives due to the similarity between the target

behaviour and other anomalous movements in the sensor data. **d** Estimated distribution of behaviours based on the 116 hours of acceleration data collected. **e** Frames taken from video captured using AloA of a black-tailed gull catching an insect in mid-air while flying over the ocean. **f** Frames taken from video captured using AloA of a black-tailed gull plucking an insect off the ocean surface.

REVIEWERS' COMMENTS:

Reviewer #1 (Remarks to the Author):

The main point of interest is that this study represents the first time machine learning has been used onboard bio-loggers to drive a data collection regime. The shearwater case study provides a nice example of where this can provide context to records of at-sea behaviour. Nonetheless, the limitations in overall battery life, and, in the case of acceleration data, the performance of the detection algorithms (identifying an estimated 50% of foraging behaviour) mean there is still some way to go before this becomes a technique that is likely to be employed in a wide range of systems. I'm therefore not sure this represents a step-change relative to simple thresholds that are already widely used to drive sampling regimes.

Nonetheless, the revisions have improved what was already a solid manuscript and this is the first in what will no doubt be a range of studies addressing the need to optimize the use of high power sensors.

Reviewer #2 (Remarks to the Author):

All of my concerns with the original manuscript have been clearly and satisfactorily addressed in the revised version. I maintain my opinion that the article is novel, timely, and of interest to bio-loggers and those that study animal movement.